# Multimodal Function of Mesenchymal Stem Cells in Psoriasis Treatment

**DOI:** 10.3390/biom15050737

**Published:** 2025-05-19

**Authors:** Jiaxin Ou, Ziqing Li, Danni Yao, Chuanjian Lu, Xiang Zeng

**Affiliations:** 1The Second Clinical School of Guangzhou University of Chinese Medicine/The Second Affiliated Hospital of Guangzhou University of Chinese Medicine/Guangdong Provincial Hospital of Chinese Medicine/Guangdong Provincial Academy of Chinese Medical Sciences, Guangzhou 510120, China; oujiaxin@gzucm.edu.cn (J.O.); 20211120792@stu.gzucm.edu.cn (Z.L.); yaodanni@gzucm.edu.cn (D.Y.); 2State Key Laboratory of Dampness Syndrome of Chinese Medicine, Guangzhou 510120, China; 3Guangdong-Hong Kong-Macau Joint Lab on Chinese Medicine and Immune Disease Research and Guangdong Provincial Key Laboratory of Clinical Research on Chinese Medicine Syndrome, Guangzhou 510120, China; 4National Institute of Stem Cell Clinical Research, Guangdong Hospital of Chinese Medicine, Guangzhou 510120, China; 5Lab of Stem Cell Biology and Innovative Research of Chinese Medicine, Guangdong Provincial Hospital of Chinese Medicine/Guangdong Academy of Chinese Medicine, Guangzhou 510120, China; 6Guangdong Provincial Key Laboratory of Chinese Medicine for Prevention and Treatment of Refractory Chronic Diseases, Guangzhou 510120, China

**Keywords:** psoriasis, mesenchymal stem cells, homeostasis, cell therapy, multimodal function

## Abstract

Psoriasis is a chronic inflammatory disease mediated by the innate and adaptive immune systems, and its pathogenesis involves multiple aspects, including abnormal interleukin (IL)-23–Th17 axis, dysfunction of Tregs and other immune cells, and a complex relationship between keratinocytes and the vascular endothelium. Dysfunction of mesenchymal stem cells in psoriatic skin may also be the main reason for the dysregulated inflammatory response. Mesenchymal stem cells, a type of adult stem cells with multidifferentiation potential, are involved in the regulation of multiple links and targets in the pathogenesis of psoriasis. Thus, a detailed exploration of these mechanisms may lead to the development of new therapeutic strategies for the treatment of psoriasis. In this paper, the role of mesenchymal stem cells in skin homeostasis, the pathogenesis of psoriasis, and the multimodal function of using mesenchymal stem cells in the treatment of psoriasis are reviewed.

## 1. Introduction

Psoriasis is a systemic chronic inflammatory skin disease mediated by the innate and adaptive immune systems and characterized by inflammatory infiltrates, acanthosis epidermidis, and hyperkeratosis [1,2,3]. Psoriasis prevalence varies by country [2], with more than 60 million adults and children being affected worldwide [4]. It is clinically characterized by erythema, scaling, and papules on the face, head, hands, feet, and back and is associated with comorbidities such as psoriatic arthritis (PSA), metabolic syndrome, depression, and cardiovascular disease [5]. It affects not only physical health, but also mental health. Studies have shown that patients with psoriasis are more likely to suffer from mental illnesses, such as depression, and even have suicidal tendencies [6,7]. Therefore, alleviating the global burden of psoriasis is an essential task [7]. Despite substantial advancement in targeted biological inhibitors used for controlling psoriasis, such as tumor necrosis factor α (TNF-α) inhibitors (etanercept, adalimumab, certolizumab, and infliximab), interleukin (IL)-12 and IL-13 inhibitors (ustekinumab), IL-17 inhibitors (secukinumab, ixekizumab, bimekizumab, and brodalumab), and inhibitors of the p19 subunit of IL-23 (guselkumab, tildrakizumab, risankizumab, and mirikizumab) [8], long-term disease modification and cure remain significant medical issues [9].

Mesenchymal stem cells (MSCs) are multipotent adult stem cells with the potential to proliferate, self-regenerate, and differentiate into multiple cell lineages [10]. They can be obtained from various sources, such as bone marrow (BM-MSCs), adipose tissue (AT-MSCs), umbilical cord (UC-MSCs), placenta, or dental pulp [11]. In recent years, studies have shown that MSCs are involved in the regulation and improvement of multiple links and targets in the pathogenesis of psoriasis and represent a multidimensional potential treatment for psoriasis. For example, MSCs have immunomodulatory and anti-inflammatory effects, especially in inhibiting the proliferation of activated T cells [12,13], suppressing dendritic cell (DC) maturation, and promoting pro-inflammatory factor secretion [14,15]. At the same time, the number of case reports and clinical trials on the application of MSCs in the treatment of psoriasis has been increasing, mostly showing good safety and promising results. Therefore, based on the first-hand knowledge obtained by our team from basic research and the first reported clinical trial (34988162), this review includes the following points: (1) the stratified pathogenesis of psoriasis; (2) the multi-dimensional role of mesenchymal stem cells in skin homeostasis; (3) the multimodal mechanisms by which mesenchymal stem cells target psoriasis; and (4) the advantages and prospects of using mesenchymal stem cells to treat psoriasis. It aims to provide an understanding of the multimodal functions of mesenchymal stem cells in the treatment of complex diseases such as psoriasis.

## 2. The Main Aspects of Psoriasis Pathogenesis

The pathogenesis of psoriasis is complex and has not been fully elucidated. However, it features a layered pathogenesis involving the interaction between innate immune cells (DCs and neutrophils), skin cells (keratinocytes, melanocytes), and adaptive immune cells (CD4+ and CD8+T cells, Th1 cells, Th17 cells, Th22 cells, Tregs) [16]. Therefore, activation of immune cells occurs in the early stage of psoriasis, especially activation of DCs and autoreactive T cells. The expansion stage includes extensive activation of T cell subsets and other immune cells and consequent pro-inflammatory and histiocytic responses.

### 2.1. Keratinocytes

In the early stage of the disease, keratinocytes produce a variety of antimicrobial peptides (AMPs) under the stimulation of injury and cytokines, which are combined with nucleic acids released by neutrophils, forming AMP–nucleic acid complexes, which break the tolerance of pDCs to their own nucleic acids, leading to activation of pDCs and production of a large amount of IFN-α [17]. Additionally, IFN-α can enhance the response of epidermal keratinocytes to IL-22 by up-regulating expression of the IL-22 receptor [18]. Keratinocytes maintain the inflammatory environment through the production of antimicrobial peptides (LL37), cytokines (IL-6, IL-1β, and TNF-α), and chemokines (e.g., CCL20, CXCL5, CXCL8, CXCL9, and CXCL10) [19]. For example, keratinocytes activate DC to produce IL-1β and IL-6 through IL-36, produce overexpressed LL37 to activate macrophages [20], regulate their own self-expression of IL-1 family cytokines, induce immune cells to produce chemokines, participate in the recruitment of neutrophils in the early stage of psoriasis [21], and further promote inflammation. The effects of the above-mentioned functions of keratinocytes promote their downstream proliferation, drive excessive proliferation of the epidermis, increase expression of angiogenesis medium and endothelial cell adhesion molecules, and lead to infiltration of immune cells into the skin lesions, resulting in a self-amplification cycle. In addition, the abnormal cell signaling and transcriptional responses of keratinocytes may be the cause of psoriasis in genetically susceptible populations [22]. Studies have found that keratinocytes activate cutaneous resident DCs and promote their production of IL-23 through key molecules of the TRAF6 (TNF receptor-associated factor 6), NF-κB, and MAPK signaling pathways, thereby promoting the development of psoriasis dermatitis [23]. STAT3 in keratinocytes is highly activated to regulate cell proliferation, inhibit cell differentiation, and promote the production of antimicrobial peptides [24], and it plays an important role in promoting the production of various inflammatory cytokines (such as IL-6, IL-17, IL-21, IL-19, IL-22, etc.) in psoriasis. Studies have found that keratinocytes overexpressing STAT3 in model mice lead to the spontaneous development of psoriasis-like lesions, and that their cytokine distribution is similar to that of human psoriasis plaques [25]. On the other hand, the specific absence of STAT3 in keratinocytes reduces psoriasis-like dermatitis [26]. NF-κB signaling is involved in the pathogenesis of psoriasis by acting on immune cells and keratinocytes, and it is highly activated in the lesioned skin of patients with psoriasis [27]. MAPK kinase is involved in the occurrence and development of psoriasis by regulating the proliferation of keratinocytes. Studies have found that activation of p38 (Phospho-p38 Mitogen-Activated Protein Kinase, p38) in the skin of mice leads to psoriasis-like dermatitis in mice. The use of p38 inhibitors can inhibit the inflammatory response induced by TNF-α or IL-17A production by keratinocytes [28,29]. In addition to the transcription factors of the above-mentioned major signaling pathways, some other transcription factors expressed in keratinocytes have become important regulatory factors of psoriasis, such as nf 2 [30], FRA195 [31], and GRHL3 [32]. Overall, the interaction between keratinocytes and immune cells (especially Th17 (T helper cell 17, Th17) leads to the induction and maintenance of psoriasis, accompanied by excessive proliferation and abnormal differentiation of keratinocytes, dilated and proliferative blood vessels, and infiltration of inflammatory cells. It is also the reason for the characteristic changes seen in psoriasis (epidermal hyperplasia, dermal vasodilation, and inflammatory cell infiltration in the dermis) [33,34,35].

### 2.2. Dysregulation of Dendritic Cell Function

Disturbance of innate and adaptive skin immune responses and non-immune cells contributes to the development and persistence of inflammation in psoriasis [36,37]. Overactivation of the adaptive immune system is the core of psoriasis pathogenesis, which is mainly manifested in the immune response mediated by DCs and T cells. DCs are widely distributed in the body [38], are professional antigen-presenting cells, which have the function of linking innate and acquired immunity, are the initiator of the body’s acquired immune response, and play an important role in the initial stage of psoriasis. The skin contains a complex network of DCs, including epidermal Langerhans cells (LCs), bone marrow-derived dermal cDCs, pDCs, and iDCs. Abnormally activated DCs can not only stimulate activated effector T cells or memory T cells, but can also activate primitive T cells and induce the activation and proliferation of T cells [39].

pDCs originate from the bone marrow, migrate to the skin under pathological conditions [39], and generate large amounts of type I IFN in response to endosomal Toll-like receptors (TLRs), such as TLR7, by recognizing viral nucleic acids [40,41]. When pDCs incorrectly recognize their own nucleic acids, they secrete IFN-α, which initiates psoriatic inflammation [42]. For example, in the early stages of the disease, keratinocytes release various AMPs and nucleic acids upon injury and cytokine stimulation, and pDCs subsequently produce large amounts of IFN-α in response to this release. IFN-α is a key upstream cytokine along the IL-23–IL-17 axis [17], because the release of IFN-α promotes the maturation of DCs and the differentiation of monocytes into inflammatory dendritic cells [43]. Then, mature cDCs and rapidly increasing iDCs release a large number of cytokines—such as IL-23, IL-12, and TNF-α—and strongly activate naive T cells to differentiate into the Th1, Th17, and Th22 subsets [39].

cDCs express anti-inflammatory cytokines, such as IL-10 and transforming growth factor (TGF)β, and promote homeostasis of Tregs under normal conditions [44]. In psoriasis, their role in maintaining immune tolerance is dysregulated [45], as cDCs are activated under the stimulation of TNF-α, IFN-α, and the LL37 RNA complex, producing a large number of inflammatory cytokines, such as IL-12 and IL-23 [39]. Normally, there are few iDCs in healthy tissue, but the number of iDCs increases significantly after an inflammatory injury. In psoriasis, TNF-α and inducible nitric oxide synthase (iNOS)-producing DCs (Tip-DCs) and 6-sulfoLacNAc DCs (slan-DCs) have been described as iDCs. In addition to the ability of iDCs to polarize T cells into the Th1, Th2, and Th17 subsets, they can induce T cells to secrete IL-17, IL-22, and TNF-α [39]. In recent years, an increasing number of studies have shown that LCs produce IL-23 and are closely related to abnormal T cells, suggesting that LCs might be involved in the pathogenesis of psoriasis vulgaris, similarly to other DCs [39]. Furthermore, LCs may be associated with psoriasis recurrence [46]. Conversely, there is also evidence for a negative regulatory role of LCs. It is suggested that the functions of LCs in psoriasis vary, and further experiments are required to fully analyze the potential role of LCs.

### 2.3. Abnormal Activation of T Cells and Key Cytokines in the Inflammatory Cycle

The inflammatory cycle of psoriasis involves T cells, such as CD8+, Th1, Th17, and Th22 cells [47]. When dermal DCs mature and activate, they release cytokines, such as IL-12 and IL-23, which, in turn, promote the occurrence and development of Th1, Th17, and Th22 differentiation. Among them, IL-23 and Th17 responses are important drivers of psoriasis development.

IL-23 is a heterodimer protein composed of the p19 and p40 subunits linked by disulfide bonds, and the p40-encoding gene has been found to be associated with psoriasis [34]. Compared with non-diseased skin, the expression of IL-23 in psoriasis lesions is significantly increased [48]. Dermal CD11c+ immune cells, including different subsets of macrophages and medullary dendritic cells [1], are the main cellular source of IL-23 in psoriasis lesions. In psoriasis, IL-23 is a key regulatory factor for Th17 differentiation in patients with psoriasis. It maintains the stability of the pathogenic phenotype of Th17 secreting cytokines [49,50] and promotes the production of IL-17 by effector and memory T cells. Furthermore, Th17 secrete IL-22, IL-26, and IL-29. The concentrations of these substances in psoriasis lesions are relatively high and amplify the inflammatory response. Evidently, IL-23 plays an important role in human psoriasis skin, and its overexpression leads to psoriasis lesions [48]. An anti-IL-23 monoclonal antibody shows a good therapeutic effect in the treatment of psoriasis; this indicates its key role [51].

The IL-17 family, consisting of six members (IL-17A, B, C, D, E, and F), plays an important role in the defense of the epithelial barrier against bacterial and fungal infections, and, when overproduced, leads to chronic inflammation and autoimmune disease [52]. In psoriasis, neutrophils, mast cells, NK cells, macrophages, and B cells can all produce IL-17, but Th17 cells are the main source of IL-17 production [53], mainly through the JAK-STAT pathway. The expression of IL-17 in psoriatic skin lesions [54] and elevated serum levels [55] are typical features of psoriasis. Additionally, antibodies against IL-17 and IL-17 receptors have been used in the treatment of psoriasis and achieved significant efficacy [56], further revealing the important role of IL-17 mainly produced by Th17 cells in psoriasis.

TNF-α can be produced by various cells, such as DCs, macrophages, and keratinocytes. It is a pro-inflammatory cytokine that plays an important role in acute and chronic inflammation and is also an important factor in the occurrence and development of psoriasis [57]. It has broad biological effects by increasing the expression of other pro-inflammatory cytokines and neutrophil chemokines, inducing vascular adhesion molecules to promote the influx of inflammatory cells, and amplifying the effects of other cytokines, including IL-17 [35,58]. Serum TNFα levels may correlate with psoriasis activity [59]. High levels of TNFα and its receptors TNFR1 and TNFR2 have been detected in psoriatic tissues. Furthermore, it can cooperate with IL-17 to increase neutrophil recruitment [60].

IL-12 promotes a Th1-type response, characterized by the production of IFN-γ, and is also considered to be crucial in the pathogenesis of psoriasis [61]. IL-22 expression is increased in psoriatic skin lesions, and its serum levels correlate with disease activity [62]. Additionally, IL-22 promotes keratinocytes to release chemokines and AMP, induces their migration, increases epidermal thickness, and inhibits their differentiation [63]. In addition to the above-mentioned cytokines, IFN-γ, anti-inflammation molecules, and signaling-regulating molecules also participate in psoriasis pathogenesis. The dysregulation of the above-mentioned key cytokines in psoriasis helps to explain the psoriasis interaction and maintenance circuits of inflammation.

Psoriasis is a systemic chronic inflammatory disease mediated by multiple systems and is easy to recur, which seriously affects the quality of patients’ lives. Current treatments for psoriasis include topical medications (e.g., salicylic acid, coal tar, anthralin, corticosteroids, vitamin D3 analogs, calcineurin inhibitors, and tazarotene for mild disease), ultraviolet (UV) therapy (for moderate disease), systemic agents (e.g., methotrexate and cyclosporine), and biologics for more severe disease. Most of the treatments play a therapeutic role by intervening in one part of psoriasis pathogenesis (a single target). For example, vitamin D3 derivatives inhibit the proliferation of keratinocytes by binding to the vitamin D3 receptor, and methotrexate prevents DNA synthesis when epidermal cells proliferate. Although immune targeting of IL-17 and IL-23 cytokines by biological agent therapy currently shows significant efficacy, this efficacy is insufficient, and the treatment lacks clinical response in individuals and fails to prevent the recurrence of psoriasis [64]. Furthermore, long-term use of biological agents may lead to adverse events, a key problem that needs to be overcome in the conventional treatment of psoriasis [65]. All in all, the traditional therapies and the advanced biological agents mentioned above may only inhibit the activity of pathogenic immune cells, but they cannot eliminate these abnormal cells [8]. Therefore, after stopping treatment, these pathogenic immune cells are reactivated and cause inflammatory lesions, leading to recurrence of the disease [35]. A more effective regimen remains an unmet medical need.

## 3. Multidimensional Functions of Mesenchymal Stem Cells in Skin Homeostasis and Psoriasis Development

### 3.1. The Essential Role of MSCs in Skin Homeostasis

MSCs, as adult stem cells with self-renewal and multidirectional differentiation potential [66], maintain their cytokine secretion balance through intercellular contact, autocrine/paracrine, and differentiation ability [67,68,69,70], such as regulating the secretion of various growth factors, adhesion molecules, chemokines, and anti-inflammatory cytokines. Under normal circumstances, MSCs in the dermis mediate innate and adaptive immune responses and play a key role in maintaining skin homeostasis (Figure 1). MSC functions include but are not limited to (1) promoting skin repair, (2) modulating immune cells, (3) influencing angiogenesis, and (4) inducing KC proliferation. In general, MSCs have (1) a specific differentiation ability in the tissue microenvironment and can differentiate into skin tissue cells at the site of skin injury to participate in injury repair [71], promoting the growth and development of skin tissue. MSCs also (2) inhibit the proliferation and migration of T cells (such as inhibiting differentiation into Th1 and Th17 cells), promote the development of Tregs [72], regulate B cells and natural killer (NK) cells [13,73], regulate the polarization of macrophages and promote their transformation into M2 type to exert anti-inflammatory and immunosuppressive effects [74,75], inhibit the secretion of TNF-α by type I mature DCs, increase the secretion of IL-10 by type II mature DCs [76], and promote the migration and phagocytosis of neutrophils through the secretion of IL-6, IL-8, and granulocyte-macrophage colony-stimulating factor (GM-CSF) [77]. The resultant secreted cytokines also (3) maintain vascular stability and promote angiogenesis, as mediated by adhesion molecules and chemokines [78,79]. These effectors are the key interstitial molecules responsive for subtle regulation of tissue homeostasis under physiological conditions (Figure 1).

### 3.2. Pathological Changes of Resident MSCs in Psoriatic Skin Lesions

Abnormal changes are found in resident MSCs in psoriasis, including changes in the expression level of their own surface molecules or abnormal secretion of cytokines, a limited ability to promote the proliferation of keratinocytes and apoptosis, and changes in immune regulation ability, such as the ability to inhibit the proliferation of T cells and the expression of pro-inflammatory factors, promote secretion of anti-inflammatory factors, inflammation, and angiogenesis, and resist oxidation. These changes destroy the balance of the skin environment in patients with psoriasis (Figure 2). However, whether resident MSCs are the victims or the initial cause of psoriasis is worth exploring.

#### 3.2.1. Abnormal Interaction with Psoriatic Keratinocytes

Recent studies found that the resident MSCs participate in the development of psoriatic skin lesions. For example, through changes in expression of their own surface molecules (such as increased levels of CD29, CD44, CD73, CD90, and CD105 and limited expression of CD34, CD45, and HLA-DR) [80] MSCs secrete various growth factors, adhesion factors, chemokines, and anti-inflammatory cytokines to change the skin microenvironment [81,82,83]. This regulates the PI3KAKT signaling pathway [84] and reduces cell-to-cell junctions [85], thus affecting keratinocytes, promoting their proliferation, limiting their apoptotic capacity, and aggravating the inflammatory response. Another study found that psoriatic keratinocytes exhibit higher potency in metabolic reprogramming (increased glycolysis and mitochondrial metabolism) and stimulation of dermal MSC proliferation (increased expression levels of stem cell factors and epidermal growth factors) [86]. Additionally, psoriatic dermal MSCs induce keratinocytes to express C3 [87], which may further contribute to the pathogenesis of psoriasis. It is therefore suggested that mutual interaction between keratinocytes and the resident MSCs may lead to escalation of lesion development.

#### 3.2.2. Changes in Immunomodulatory Capacity

MSCs in psoriatic skin lesions have altered immunomodulatory capacity compared with that in normal human skin, including reduced ability to inhibit T cell proliferation [88], increased expression of pro-inflammatory factors, and decreased secretion of anti-inflammatory factors [89], such as decreased levels of ROR-γt and T-bet [90]. The decreased expression levels of ROR-γt help inhibit the ability of T cells to differentiate into Th17 cells, and T-bet is a master regulator of TH1 cell differentiation that inhibits the Th2 and Th17 transcriptional programs. Psoriatic MSCs also inhibit IL-10 function and Th17 differentiation, thereby affecting the Th17/Treg balance [91]. Moreover, expression of a pro-inflammatory *miRNA (miR-155)* becomes up-regulated, and the levels of genes related to the regulation of immunity (such as *PGE2*, *IL-10*, and *TLR4*) become down-regulated [92]. Hence, the abnormality of MSCs derived from psoriatic skin lesions and the changes in their immunoregulatory ability lead to a local increase in T cells in the skin lesions and promote the occurrence and development of local inflammatory responses.

#### 3.2.3. Pro-Angiogenesis and Pro-Inflammatory Mediators Increased

One of the characteristic changes of psoriasis is prominent dermal vasodilation, and vascular endothelial growth factor plays an important role in stimulating angiogenesis and psoriasis pathogenesis [33,93,94,95]. Dermal-derived MSCs from patients with psoriasis have abnormal proliferative capacity and pro-inflammatory and pro-angiogenic potential, which may be involved in the early development of psoriasis. Comparison of dermal MSCs between psoriasis patients and normal healthy people has shown that the expression of pro-inflammatory and angiogenesis-related mediators of MSCs in psoriasis patients is increased, such as lipopolysaccharide (LPS)-induced TNFα, transcription factor (LITAF), and vascular endothelial growth factor [96,97,98,99], Additionally, there is increased expression of angiogenic genes, such as *TGF*-β and the angiopoietin gene. The expression of *miR-155*, a pro-inflammatory factor that plays an important role in the inflammatory response of psoriasis, is also significantly increased [100,101]. Moreover, enhanced expression of factors associated with pro-inflammatory phenotypes [102] and abnormal glucose metabolism [103] lead to local vascular abnormalities in patients with psoriasis.

#### 3.2.4. Decreased Antioxidant Capacity

One study found that the highest content of iNOS in psoriatic skin lesions affects the antioxidant capacity of MSCs, thereby mediating the development of psoriatic skin lesions [104]. Furthermore, Sah et al. found that, in mice with imiquimod-induced psoriasis, MSCs in skin-like lesions had a reduced ability to utilize extracellular superoxide dismutase [105], decreasing their antioxidative ability.

## 4. The Therapeutic Effect of MSCs in Psoriasis

Cumulative evidence suggest that MSCs play multimodal roles in the treatment of psoriasis through a variety of functions. Therefore, the significance and persistence of therapeutic efficacy and the extension of rash relapse time are closely related to the participation of MSCs in the regulation of multiple links in the pathogenesis of psoriasis through cell-to-cell contact, paracrine/autocrine signaling, or differentiation ability, thus resulting in a synergistic effect that is different from that of conventional psoriasis therapies aiming at a relatively single target. However, MSCs are not a type of homogeneous cells, but rather are a group of heterogenous cells with discernible phenotypic and functional diversity. Additionally, MSCs are present in almost all organs but are mainly obtained from bone marrow (BM-MSCs), the umbilical cord (UC-MSCs), placental tissue, dental pulp, and adipose tissue (AD-MSCs). Depending on the source of MSCs, their biological properties vary, including differentiation capacity, paracrine potential, and immunomodulatory properties. For example, ADSCs have stronger inhibitory effects on peripheral blood B cells, T cells, and NK cells in vitro than BM-MSCs and UC-MSCs, but all three can promote regulatory T cell and Th1 cell polarization [106]. The immunomodulatory effects of MSCs are exerted by interactions with immune cells through cell-to-cell contacts and paracrine signaling, involving T cells, B cells, NK cells, macrophages, monocytes, dendritic cells [99,107], and neutrophils [12,13]. Therefore, we summarize the effectiveness of MSCs derived from different tissue sources in the treatment of psoriasis in preclinical (Table 1) and clinical studies (Table 2).

### 4.1. Adipose-Derived Mesenchymal Stem Cells

#### 4.1.1. Preclinical Studies

AD-MSCs have immunomodulatory potential in the treatment of psoriasis by inhibiting the production of Th17-related cytokines, such as IL-17A and TNF-α, which alleviates imiquimod (IMQ)-induced skin pathological changes associated with psoriasis in mice [108]. Similar results were obtained by Feng et al. [109], although they also found that AD-MSCs improved psoriasis by negatively regulating ROS, a pro-inflammatory factor that can induce the release of other inflammatory factors and oxidative damage in psoriasis. It is suggested that AD-MSCs also have the potential for antioxidative regulation in the treatment of psoriasis.

#### 4.1.2. Clinical Studies

AD-MSCs in the treatment of psoriasis not only are safe and have durable efficacy, but also show superior potential in delaying the recurrence of psoriasis. De Jesus et al. [110] reported improvement in erythematous scales in two patients with psoriasis vulgaris who were treated with AD-MSCs, maintaining PASI-50 for 9.7 months. We were the first to report a clinical trial of the results of AD-MSCs infusion in patients with moderate to severe psoriasis [111]. Our data suggested that four of the seven included patients completed the trial, and two completed the 1-year follow-up, achieving and maintaining PASI-50 after 1 year of no treatment. Even during the follow-up period, one patient maintained PASI-50 for nearly 3 years.

### 4.2. Human Umbilical Cord Mesenchymal Stem Cells (hUC-MSCs)

#### 4.2.1. Preclinical Studies

In animal experiments, hUC-MSCs effectively treat psoriasis by regulating multiple pathways, including the expression of inflammatory mediators, the balance of Th1/Th2/Th17 cells, the penetration of immune cells into the skin, and the JAK-STAT pathway [15,112]. Particularly, Sah SK et al. [105] found that SOD3-transduced MSCs regulate the infiltration and function of immune cells, especially DCs, neutrophils, and Th17 cells, regulate epidermal dysfunction, and inhibit disease progression to a greater extent than non-transduced MSCs. Lee YS et al. [113] used two different methods to construct a psoriasis rash model (IMQ-induced and IL-23-mediated) and different intervention time points and proved that UC-MSCs can treat developing psoriasis and prevent psoriasis. Additionally, they also found that hUC-MSCs not only directly regulate the activation and differentiation of CD4+ T cells, but also indirectly regulate the activation and differentiation of CD4+ T cells by regulating dendritic cell function. Lin et al. [114] compared the improvement effect of different concentrations of hUC-MSCs on psoriasis-like skin lesions in *BALB/c* mice and found that high concentrations of hUC-MSCs had the best therapeutic effect. Chen et al. [115] demonstrated that subcutaneous injection was a better injection method than tail vein injection (associated with pulmonary embolism), and hUC-MSCs suppressed psoriatic skin inflammation by inhibiting interleukin-17-producing γδT cells.

#### 4.2.2. Clinical Studies

Chen H et al. [116] reported two patients with psoriasis vulgaris who maintained PASI-50 for 4–5 years after receiving umbilical cord MSC therapy. Ahn H et al. [117] reported a 47-year-old patient who received 25 years of conventional treatment for psoriasis without significant improvement. The patient’s erythema gradually disappeared after receiving three rounds of minimal MSCs via vein transplantation and local transplantation within 2 weeks. Three months after the first round of treatment, the PASI score decreased from 9.9 to 1.7, the DLQI score decreased from 27 to 3, and the quality of life was significantly improved. Psoriasis also did not recur. Additionally, there were no reports of adverse reactions or side effects. Cheng L et al. [118] speculated that Treg levels can be used as an effective biomarker to predict the clinical efficacy of embryonic MSC transplantation, in addition to being used for efficacy evaluation.

### 4.3. Human Embryonic Mesenchymal Stem Cells (hE-MSCs)

#### 4.3.1. Preclinical Studies

Kim CH et al. [119] established a mouse model of IMQ-induced psoriasis and found that hE-MSCs inhibited the expression of Th17-cell-related factors (IL-17 and IL-23) and implemented immunomodulation to psoriasis, which may be a key factor in improving psoriasis.

#### 4.3.2. Clinical Studies

According to a literature search, to date, no reports have been published.

### 4.4. Gingival Mesenchymal Stem Cells (GMSCs)

#### 4.4.1. Preclinical Studies

Ye et al. [120] found that GMSCs could significantly improve IMQ-induced psoriatic skin inflammation in mice, reduce Th1- and Th17-related cytokines, including IFN-γ, TNF-α, IL-6, IL-17A, IL-17F, IL-21, and IL-22, up-regulate the percentage of spleen CD25+CD3+ T cells, and down-regulate the percentage of spleen IL-17+CD3+ T cells.

#### 4.4.2. Clinical Studies

Wang SG et al. [121] reported a 5-year history of severe plaque psoriasis persisting after multiple local and systemic treatments. After receiving 2 consecutive weeks of allogeneic GMSC infusions (3 × 10^6^/kg), a gradual clearing of the rash plaques was observed. Five weeks later, an additional 3-weekly MSC infusion was performed, and 1 week after the last injection, the patient’s psoriatic lesions completely disappeared. Additionally, the patient was followed up for 3 years without recurrence.

### 4.5. Human Amniotic Membrane-Derived Mesenchymal Stem Cells (hAMSC)

#### 4.5.1. Preclinical Studies

Imai et al. [122] found that hAMSCs significantly reduced IL-17A and IL-22 production by low T cells in IMQ-induced rash. Furthermore, co-culture of normal human epidermal keratinocytes (NHEKs) with hAMSCs inhibited the production of IFNγ- or TNFα-induced IL-8 by human epidermal keratinocytes. Yang et al. [123] showed that topical administration of human amniotic epithelial cell (hAEC)-derived AEC-SCs effectively ameliorated IMQ-induced psoriasis-like skin lesions and skin inflammation, in which IL-1ra played an important role. This provides a new therapeutic method and strategy for MSCs to treat psoriasis.

#### 4.5.2. Clinical Studies

Following an extensive search, no reports have yet been published on this topic.

### 4.6. Human Tonsil-Derived Mesenchymal Stem Cells (T-MSC)

#### 4.6.1. Preclinical Studies

Kim et al. [124] showed that the PD-1/PD-L1 pathway may be a key factor in regulating Th17 immune responses in psoriasis [125,126]. Additionally, T-MSCs migrate into injured inflammatory tissue in a mouse model of liver injury. Therefore, it is hypothesized that T-MSCs might migrate to skin lesions and modulate immune responses through intercellular and paracrine signaling. Kim et al. demonstrated that T-MSCs effectively inhibited Th17 responses in a PD-L1-dependent manner in a mouse model of psoriasis.

#### 4.6.2. Clinical Studies

To date, no reports on this topic have been published.

### 4.7. Summary of MSC-Based Psoriasis Treatments

Basic (Table 1) and/or clinical studies (Table 2) suggested that the application of MSCs with different origins demonstrated a certain degree of effectiveness in controlling the occurrence and development of psoriasis, with great therapeutic potential. In clinical application and follow-up of MSCs from several sources, there are no reports of adverse events or serious adverse events. The maintenance of the PASI-50 achievement rate is 1–3 years (MSCs derived from the umbilical cord even reach 4–5 years), which significantly delays the recurrence of psoriatic rash. However, the above-presented clinical trials have limitations, including insufficient sample size, limited follow-up period, no control group, and no blinding. In the future, the sample size should be expanded to conduct randomized controlled trials to improve the rigor, quality, and reliability of clinical data conclusions. In terms of basic research, in vivo research in mice is currently the main animal model. Most modeling methods are the IMQ-induced psoriasis rash model, and hMSCs from various sources are involved. Studies have shown that MSCs decrease and inhibit the occurrence and development of psoriasis through multiple pathways, including anti-inflammatory effects and immunomodulatory effects. MSCs regulate DCs in psoriasis [113]. Although there is little relevant evidence at present, according to previous research [127,128,129,130,131], MSCs exert effects related to DCs, such as inhibiting maturation of DCs (typically reduced expression of DC antigens) and their ability to promote proliferation and differentiation of CD4+ T cells. Moreover, one of the main aspects of psoriasis pathogenesis is abnormal activation of T cells caused by DC dysregulation. For this reason, we can hypothesize that MSCs targeting pathogenic DC subsets are another effective strategy for the treatment of psoriasis.

To sum up, due to the different experimental designs and endpoints of each study, more studies in different directions are needed to further clarify the clinical and scientific significance of MSCs in the treatment of psoriasis.

### 4.8. MSCs Targeting Pathogenic DCs

Notably, the modulatory effect of MSCS on dendritic cells (DCs) in psoriasis treatment is remarkable. Dendritic cells (DCs) are important antigen-presenting cells that play a key role in the innate and adaptive immune systems. Although the pathogenesis of psoriasis is still unclear, a large number of studies have suggested that the abnormal activation of T cells caused by dendritic cell dysregulation is one of the main elements of the pathogenesis of psoriasis. At the same time, regulation of DC maturation and related cytokines produced by DCs has shown certain efficacy. Previous studies have suggested that MSC therapy for psoriasis may be related to dendritic cells [113]. In addition, other experimental studies have shown similar results, that MSCs inhibit DC maturation by modulating phosphorylation of the STAT pathway in uveitis [132]; MSCs can inhibit the differentiation and maturation of DCs by secreting soluble factors or by direct contact, reduce the expression of CD86, CD80, CD40, and HLA-DR [76,127,128,129,130,131,133,134,135], and decrease their ability to induce T cell proliferation [136,137]; MSCs may inhibit DCs by up-regulating the secretion of IL-10 [138,139]; in addition, MSCs could down-regulate the expression of IL-12 and IFN-α in mature DCs [76,134,135]. Therefore, MSC therapy may play a major immunomodulatory role by regulating dendritic cells (Figure 3). Factoring in that pathogenic DCs are the upstream players in psoriasis development, targeting of local DC subsets by MSCs might be the underlying mechanism for their efficacy in the treatment of psoriasis, in particular the control of disease recurrence. However, since pathogenic DCs reside in skin, whether systemic delivery confers local “homing” of MSCs is an intriguing question.

## 5. Conclusions and Perspective

Psoriasis is a multisystem, systemic, and chronic inflammatory disease. MSCs are adult stem cells with multidirectional differentiation potential, which have the potential to proliferate, self-regenerate, and differentiate into multiple cell lineages. For the treatment of psoriasis, MSCs are involved in the regulation and improvement of multiple links and targets and are a potential multidimensional treatment. Additionally, the abnormality of MSCs also affects psoriasis pathogenesis and plays an important role in maintaining skin homeostasis.

Undoubtedly, MSCs have great potential in the treatment of psoriasis. According to current research reports, the most worrying aspect of MSC treatment might be tumor-related adverse events. Although there are current reports suggesting that MSCs may support the occurrence and development of tumors by interacting with tumor cells through paracrine means or by direct cellular contact [140], it is particularly important and cannot be ignored that a larger number of studies have reported their anti-tumor properties [141,142,143,144]. Nonetheless, to our best knowledge, among thousands of clinical studies, there is no one report provided unequivocal evidence of tumor formation after MSC transplantation. In addition, this article focuses on treatment with MSCs in the field of psoriasis. However, according to the current research, no adverse events related to tumors have been reported, and most feedback on the safety of its treatment has been provided [117,118]. Indeed, the exact function of MSCs in treatment remains controversial, especially regarding the dual impact on cancer cells. We should conduct further in-depth research to clarify the mechanisms involved and address these issues, such as by modifying MSCs to transform them into undisputed therapeutic tools [145,146]. Although MSCs are associated with supporting the occurrence of tumors, the evidence for an anti-tumor effect of MSCs is compelling, and their significant role discovered in immune system therapy may eventually lead to new targeted therapies. For this reason, we should view the pros and cons of MSC treatment with a scientific and rigorous attitude, conduct more research on the therapeutic mechanism of MSCs, and better understand these complex molecular mechanisms to ensure minimal side effects and little or no risk of tumor cell growth during clinical application.

Future preclinical research requires the development of new technologies to track the distribution, mode of action, metabolism, and persistence of MSCs in vivo. Furthermore, exploration of other possible targets and main mechanisms of action of MSCs in the treatment of psoriasis are required. In terms of clinical trials, it is necessary to continuously improve the clinical design, including optimizing the therapeutic dose, dosing interval, and dosing frequency, expanding the sample size, setting a control group, and using blinding methods to further clarify MSC safety and effectiveness. Especially in terms of safety, the current reports of adverse events related to the use of MSCs are not comprehensive enough. Safety should be guaranteed on the premise of ensuring effectiveness. This will have far-reaching clinical and scientific significance for the treatment of psoriasis.

## Figures and Tables

**Figure 1 biomolecules-15-00737-f001:**
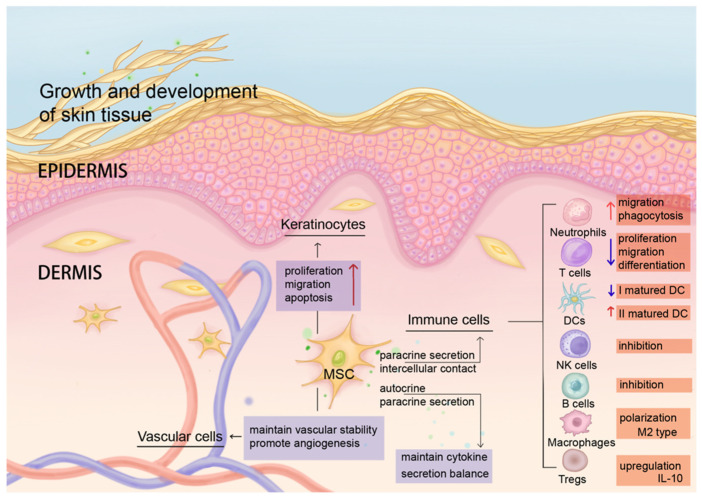
MSCs maintain homeostasis of the skin microenvironment and coordinate anti-inflammatory and regenerative immune regulation. Under normal circumstances, resident MSCs in human dermis maintain skin homeostasis through intercellular contact and paracrine secretion. MSCs regulate the functions of immune cells in terms of their migration, differentiation, inhibition, cytokine expression, phagocytosis, etc. (red arrows indicate up-regulation of function, purple arrows indicate down-regulation of function). MSCs also affect the proliferation, migration, and apoptosis of keratinocytes, maintain vascular stability, and promote angiogenesis.

**Figure 2 biomolecules-15-00737-f002:**
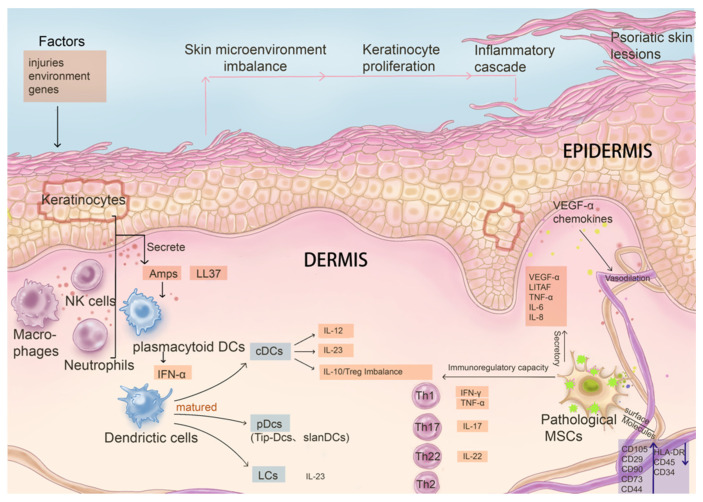
The influence on and interaction relationship of pathological mesenchymal stem cells with key immune system cells and factors in the pathogenesis of psoriasis. In psoriasis, the phenotype and function of MSCs change significantly following disease progression, contributing to disruption of skin homeostasis. Pathological MSCs have limited capacity in maintaining the balance of proliferation and apoptosis of keratinocytes, as their secretion function is altered. In addition, immune regulatory capacity is changed in abnormal MSCs, leading to pathological cascades such as abnormal maturation of DCs stimulated by AMPs and LL37, abnormal differentiation of Th cells, inhibition of T cell proliferation, and down-regulation of the expression of pro-inflammatory factors. Additionally, pathological MSCs impact angiogenesis ability as well.

**Figure 3 biomolecules-15-00737-f003:**
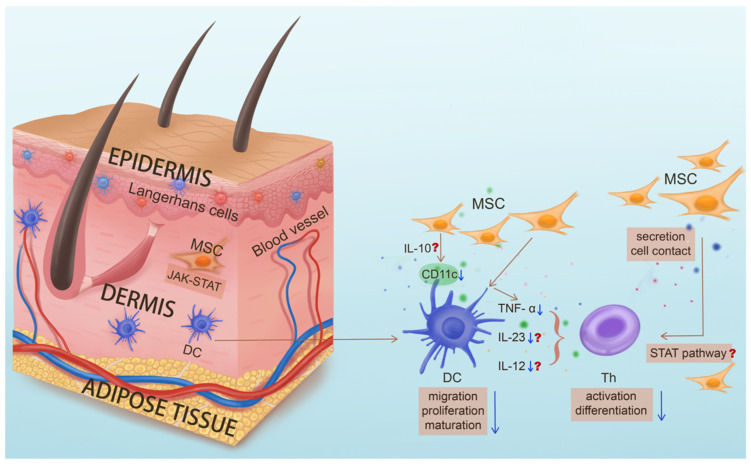
The key role of dendritic cells in the pathogenesis of psoriasis and the potential functions of mesenchymal stem cells targeting pathogenic dendritic cells. In psoriasis, upstream dendritic cells (DCs) abnormally mature and activate, secreting key inflammatory factors such as IL-23 to promote the differentiation of Th cells and mediate the downstream inflammatory response. MSC treatment for psoriasis may inhibit differentiation and maturation of DC through secretion of IL-10, phosphorylation of the STAT pathway, direct contact, etc. and down-regulate the expression of IL-12 and IFN-α, thereby inhibiting differentiation of T cells and their secretion of inflammatory cytokines, and thus improving the occurrence and development of psoriasis. (Question marks indicate functions of MSC that have been validated in other disease treatments but have yet to be studied in psoriasis treatment).

**Table 1 biomolecules-15-00737-t001:** Comparison of specific information of mesenchymal stem cells from different sources in preclinical studies.

Type of Cell	Animal Model (Sex, Age)	Disease Model	Injection Method	Treatment	Effectiveness	Follow-Up	Reference
AD-MSCs	——	IMQ-induced psoriasis-like inflammation	intravenous injection	On days 0, 2, and 4, either 1.0 × 10^6^ cells or PBS were injected into IMQ-treated skin	Inhibits the production of Th17-related cytokines (such as IL-17A and TNFα) and minimizes IMQ-induced skin changes in psoriasis	6 days	[78]
hAD-MSCs	C57BL/6 mice (male, 8–10 weeks)	IMQ-induced psoriasis-like inflammation (every day for 7 days)	subcutaneous injection	On days 1 and 4, 1.0 × 10^6^ hAD-MSCs were injected. Control mice received an equal volume of PBS.	hAD-MSCs might treat inflammatory disease by suppressing ROS production	8 days	[79]
UC-MSCs	C57BL/6 mice (male, 6 weeks)	IMQ-induced psoriasis-like inflammation	subcutaneous injection	On days 0 and 6, the control group and the model group received a 200 μL injection of normal saline, and the hUC-MSCs group received an injection of 2.0 × 10^6^ cells.	hUC MSCs may play an immunomodulatory role by regulating IL-6 and alleviating the symptoms of psoriasis	12 days	[85]
hUCB-MSCs	adult albino rats (male, —)	IMQ-induced psoriasis-like inflammation (for 12 consecutive days)	subcutaneous injection	On day 6, the rats received an MSC injection (2.0 × 10^6^ cells per injection in 2 mL of media) at four corners around the edge of the inflamed area of the skin.	The erythema, scale, and thickening of psoriasis were significantly reduced	12 days	[83]
hUCB-MSCs	C57BL/6 mice (female, 8–10 weeks)	IMQ-induced psoriasis-like inflammation (every day for 6 days)	intravenous injection	1.5 × 10^6^ cells were injected through the tail vein on the first and fourth days	It alleviated psoriasis-like skin inflammation in the mouse model, which may be related to the inhibition of Th17 cells and TNFα production	7 days	[84]
hUCB-MSCs	C57BL/6 mice (male, 8–12 weeks)	IL-23-mediated psoriasis-like skin inflammation in mice; IMQ-induced psoriasis-like inflammation	subcutaneous injection	IL-23 mediated model: 2.0 × 10^6^ cells were injected on days 1 and 7 after induction of psoriasis-like skin. IMQ-induced model: mice received an injection of 2.0 × 10^6^ cells every day for 5 days. Control group: an equal volume of phosphate-buffered saline was given at the same time points	Treatment inhibited pro-inflammatory cytokines (IL-6, IL-17, and TNFα) and the expression of chemokines (CCL17, CCL20, and CCL27) in mouse skin, providing prevention and treatment of psoriasis-like skin inflammation in mice	15 days	[87]
hUCB-MSCs	BALB/C mice (male, 6–8 weeks)	IMQ-induced psoriasis-like inflammation (for consecutive 7 days), plus subcutaneous injection of 5 μL of rmIL-12 and LPS once at the beginning of model creation	intravenous injection	The intervention group was divided into four intervention conditions: UC stem cells 1.0 × 10^7^/kg, 2 × 10^7^/kg, and 4 × 10^7^/kg and fresh UC stem cells 2 × 10^7^/kg	It had an obvious effect on the treatment of psoriasis in mice, and the high concentration of hUC-MSCs had the best therapeutic effect.	7 days	[88]
hUCB-MSCs	BALB/C mice (female, 6–8 weeks)	IMQ-induced psoriasis-like inflammation (for 6 consecutive days)	intravenous injection	IMQ group: 100 μL sterile PBS solution, IMQ+MSC group: 1.0 × 10^6^/cells MSC suspension in the same volume	MSCs could alleviate the severity of psoriasis-like dermatitis in mice by inhibiting TYK2 phosphorylation	8 days	[86]
hUCB-MSCs	C57BL/6 mice (—, 6–8 weeks)	IMQ-induced psoriasis-like inflammation (for 6 consecutive days)	subcutaneous injection	Subcutaneous injection with hUC-MSCs (2 × 10^6^/cells) in 200 μL PBS. Control mice received an equal volume of PBS	hUC-MSCs suppressed psoriasis-like skin inflammation by suppressing γδ T cells, specifically IL-17-producing γδ T cells	6 days	[89]
hE-MSCs	C57BL/6 mice (female, 8 weeks)	IMQ-induced psoriasis-like inflammation (for 6 consecutive days)	subcutaneous injection	On the second and fourth days after the start of local application of IMQ, 2.5 × 10^6^/cells of MSCs in 200 uL PBS were injected into mice	Inhibition of Th1- and Th17-related cytokines and IMQ-induced psoriasis skin changes	7 days	[90]
GMSCs	C57BL/6 mice (female, 8 weeks)	IMQ-induced psoriasis-like inflammation (for 7 consecutive days)	intravenous injection	2 × 10^6^ 2D-GMSCs or 3D-GMSCs in 200 μL PBS via the mouse tail vein on day 1 and day 4, with consecutive IMQ treatment. The IMQ control group received an equal volume of PBS at the same time points	The proportion of Treg cells increased, while the proportion of Th17 cells decreased, indicating that GMSCs play an immunomodulatory and anti-inflammatory role	8 days	[92]
hAMSC	C57BL/6 mice (—)	IMQ-induced psoriasis-like inflammation (for 5 consecutive days)	intravenous injection	On days 0 and 3, the MSC treatment group was injected with hAMSCs (2 × 10^5^ cells) suspended in 100 μL mouse serum. The control group was injected with 100 μL mouse serum.	hAMSCs decreased both IL-17A and IL-22 production by cutaneous γδ-low T cells in IMQ-treated skin	5 days	[93]
T-MSCs	C57BL/6 mice (female, 8 weeks)	IMQ-induced psoriasis-like inflammation (for 6 consecutive days)	intravenous injection	1 × 10^6^ T-MSCs were injected into the mouse tail vein on days 1 and 3 of the IMQ application period. Control mice received an equal volume of PBS at the same time points	T-MSCs could prevent autoimmune disease (psoriasis) by enhancing the PD-1/PD-L1 pathway	7 days	[95]

**Table 2 biomolecules-15-00737-t002:** Comparison of specific information of mesenchymal stem cells from different sources in clinical studies.

Type of Cell	Clinical Study	Number of Male/Female Patients	Age (Years)	Treatment	Safety (Treatment-Related Adverse Event)	Inclusion Criteria	Effectiveness	Follow-Up	Reference
Autologous AD-MSCs	Case report	2 (1/1)	58, 28	Two to three intravenous infusions of AD-MSCs at a dose of 0.5–3.1 × 10^6^ cells/kg	No adverse reaction was recorded in the first year of follow-up	Psoriasis vulgaris and psoriatic arthritis	PASI scores improved significantly	12 months	[76]
Allogeneic AD-MSCs	A single-center, single-arm pilot study (NCT03265613)	7 (6/1)	18–65	At weeks 0, 4, and 8, 0.5 × 10^6^ cells/kg body weight and 2–3 mL/min infusion rate were used for intravenous infusion of AD MSCs	16 adverse events, mild (level 1, 94%) or moderate (level 2, 6%). The most common adverse event was transient fever (31.2%)	Moderate to severe psoriasis vulgaris (PASI > 10 or BSA > 10%)	The PASI score decreased gradually from baseline to week 12	12 weeks	[77]
Allogeneic UC-MSCs	Case report	2 (1/1)	35, 26	Dose of 1 × 10^6^ cells at different frequencies	None	Psoriasis vulgaris	Psoriasis was relieved without recurrence	4–5 years	[80]
Allogeneic UC-MSCs	Case report	1 (1/0)	47	Three rounds of treatment were given within 2 weeks. In the first round, intravenous transplantation (3 × 10^6^ cells) and local transplantation (1 × 10^6^ cells) were given. The second and third rounds included only local transplantation, and they were conducted once a week	None	Psoriasis vulgaris	Psoriasis was improved, and there was no side effect or recurrence during the follow-up	5 months	[81]
Allogeneic UC-MSCs	A phase 1/2a, single-arm study (NCT03765957)	12	18–65	There were four groups (A, B, C, and D). Groups A and B received intravenous 1.5 × 10^6^/kg and 2 × 10^6^/kg. Groups C and D received intravenous 2.5 × 10^6^/kg and 3 × 10^6^/kg (mesenchymal stem cells at baseline and every 4 weeks, two times)	N/A	Psoriasis vulgaris	PASI improvement ≥ 75%, n = 6. One patient had improvement in PASI 90 at the 6-month follow-up and remained relapse-free for 1 year without using any other traditional drugs.	6 months	[82]
Allogeneic GMSCs	Case report	1 (1/0)	19	The dose for two consecutive weeks was 3 × 10^6^/kg for infusion treatment. After five weeks, MSC infusion was conducted three times a week	None	Severe plaque psoriasis	The plaque was gradually cleared, and no adverse reaction was reported. One week after the last injection, the psoriasis lesions completely disappeared. There was no recurrence in the three-year monitoring.	3 years	[91]
Allogeneic UC-MSCs	A phase 1/2a, single-arm study	30	18–65	Subjects in this arm received six UC-MSC infusions (each time 1 × 10^6^/kg) within 8 weeks. The first time to the fourth time were given once a week for 4 successive weeks; then, the last two times were given once every 2 weeks.	N/A	Psoriasis vulgaris	N/A	12 months	Safety and efficacy of UC-MSCs in patients with psoriasis vulgaris (NCT02491658) ClinicalTrials.gov
Allogeneic UC-MSCs	A randomized, positive controlled phase 1 trial	57	18–60	Low-dose: 1 × 10^6^ cells/kg; high-dose: 3 × 10^6^ cells/kg For the first to the fourth time, the subjects were given treatment once a week for 4 successive weeks, and the last two times were given once every 2 weeks. Active comparator: methotrexate (each time 5–25 mg orally once a week for 16 successive weeks)	N/A	Moderate and severe plaque psoriasis	N/A	52 weeks	Safety and efficacy of UC-MSCs in patients with plaque psoriasis (NCT03424629) ClinicalTrials.gov
Allogeneic AD-MSCs	A phase 1/2, single-group study	8	18–65	Drug: calcipotriol ointment (twice daily for 12 weeks) AD-MSCs: 2 × 10^6^ cells/kg	N/A	Moderate to severe psoriasis vulgaris (PASI > 10 or BSA >10%)	N/A	12 weeks	Efficacy and safety of AD-MSCs plus calcipotriol ointment in patients with moderate to severe psoriasis (NCT03392311) ClinicalTrials.gov
Allogeneic AD-MSCs	A single-group study	8	18–65	Drug: calcipotriol ointment (twice daily for 12 weeks). AD-MSCs: 2 × 10^6^ cells/kg; drug: PSORI-CM01 granule 5.5 g orally once a day for 12 weeks	N/A	Moderate to severe psoriasis (PASI > 10 or BSA >10%)	N/A	12 weeks	Efficacy and safety of AD-MSCs plus calcipotriol ointment and PSORI-CM01 granule in psoriasis patients (NCT04275024) ClinicalTrials.gov
Allogeneic AD-MSCs	A randomized controlled phase 1/2 trial	16	18–65	Gu Ben Hua Yu group: Gu Ben Hua Yu formula was orally administrated once a day for 12 weeks (except the day of the infusion of AD-MSCs) + AD-MSCs: 2 × 10^6^ cells/kg (at weeks 0, 2, 4, 6, and 8)	N/A	Psoriasis vulgaris (PASI > 7 or BSA >10%)	N/A	12 weeks	Comparison of PSORI-CM01 formula vs. Gu Ben Hua Yu formula combined with AD-MSCs in psoriasis (NCT04785027) ClinicalTrials.gov
hUCB-MSCs	A phase 1, single-center, randomized, open-label study	9	19–65	Patients were treated by FURESTEM-CD Inj. Subcutaneous injection: 5.0 × 10^7^ cells; 1.0 × 10^7^ cells; 2.0 × 10^8^ cells	N/A	Moderate to severe plaque-type psoriasis	N/A	144 weeks	Safety of FURESTEM-CD inj. in patients with moderate to severe plaque-type psoriasis (NCT02918123) ClinicalTrials.gov
Allogeneic UC-MSCs	A phase 1/2, single-group study	5	18–65	UC-MSC intravenous injection at a dose of 2.0 × 10^6^ cells at weeks 0, 2, 4, 6, and 8 for 12 weeks	N/A	Moderate to severe psoriasis vulgaris (PASI > 7 or BSA > 10%)	N/A	12 weeks	Efficacy and safety of expanded UCMSCs on patients with moderate to severe psoriasis (NCT03745417) ClinicalTrials.gov

## Data Availability

Not applicable.

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
