# Peer review of "Multimodal Function of Mesenchymal Stem Cells in Psoriasis Treatment"

_biomolecules, 2025, doi:10.3390/biom15050737_

Round 1
Reviewer 1 Report
Comments and Suggestions for Authors
Please find below some comments for the authors. Please make sure that tables are included in the manuscript:
Specific comments:
Jiaxin Ou et are trying to summarize the Functions of Mesenchymal Stem Cells in Skin Homeostasis and Psoriasis Treatment. Th review needs some changes before it is accepte for publication.
1.The review indeed includes several information about the role of MSC in psoriasi treatment but not so many regarding the skin homeostasis therefore the authors should revise the title.
2. In the Introduction section the last sentence of the the second paragraph is too long and therefore diffcicult to follow. Please reviese.
3. The section regarding keratinocytes is too short. Since this type of cells are major players in psoriasis the authors should elaborate more about their role in skin homeostasis.
4. In the cytokine section it would be useful if the authors write a short paragraph about the IL-23 cytokine.
5. In the same section the last paragraph includes information about the role of Vitamin D3 analogs in psoriasis which may not be necessary for the the present review. Furthermore, in the same paragraph there are some parentheses including numbers instead of references??
6.I believe that the paragraph 4.3 should be moved before the the paragraph 4.2
7. Please check the manuscript for grammar and syntax.
Author Response
Reviewer #1:
comments 1:The review indeed includes several information about the role of MSC in psoriasi treatment but not so many regarding the skin homeostasis therefore the authors should revise the title.
Response 1: We thank the reviewer for the suggestion and we deem it is a reasonable one. We therefore revised the tile to be “Multimodal Function of Mesenchymal Stem Cells in Psoriasis Treatment.”
comments 2:In the Introduction section the last sentence of the the second paragraph is too long and therefore diffcicult to follow. Please reviese.
Response 2: Revision has been made according to the suggestion.
comments 3:The section regarding keratinocytes is too short. Since this type of cells are major players in psoriasis the authors should elaborate more about their role in skin homeostasis.
Response 3: We agree with this comment and have made necessary supplements regarding keratinocytes in Section 2.1 and 3.2.1 in the revised manuscript.
comments 4:In the cytokine section it would be useful if the authors write a short paragraph about the IL-23 cytokine.
Response 4: We agree with this comment and have added more information regarding IL-23 in Section 2.3.
comments 5:In the same section the last paragraph includes information about the role of Vitamin D3 analogs in psoriasis which may not be necessary for the the present review. Furthermore, in the same paragraph there are some parentheses including numbers instead of references??
Response 5: We thank the reviewer for the comment. Vitamin D3 analogues as a traditional treatment in psoriasis bear the limitation of action on a single target, other than multifunctional effects from MSCs. We therefore believe this comparison is necessary.
We apologize for the appearance of some parentheses containing numbers presumably caused by the format error of reference-editing tool. We have double-check the references throughout the entire manuscript.
comments 6:I believe that the paragraph 4.3 should be moved before the the paragraph 4.2
Response 6: We thank the reviewer for the comment. Section 4.2 is a summary of Section 4.1 and therefore it goes after Section 4.1; while in Section 4.3 a specific function of MSCs on local dendritic cells was described. Considering the logic flow, we remain the structure as it was in the revision.
comments 7:Please check the manuscript for grammar and syntax.
Response 7: We thank the reviewer for the comment and suggestion. The manuscript was linguistically edited by Charlesworth Author Service before submission. However, per suggestion we have re-reviewed the manuscript and double-checked the grammar and syntax.
Reviewer 2 Report
Comments and Suggestions for Authors
Like other chronic inflammatory diseases, psoriasis severely reduces the patient's quality of life and represents a significant economic burden. This article aims to review the currently available knowledge regarding the development of psoriasis and the potential treatment strategy that involves mesenchymal stem cells. Having first-hand clinical knowledge in this area, the authors have comprehensively summarized the pre-clinical and clinical data, providing insights into future therapeutical investigations. However, several issues should be addressed in the revision.
- Tables 1 and 2 are not available for review.
- The article does not define multiple abbreviations, although they may be familiar to experts in the area.
- Figures 1 and 2 seem to have been prepared using a third-party platform. Relative copyright information should be provided.
- The caption of Figure 2 is too simple, lacking the necessary information to guide the audience.
- Accumulating reports have demonstrated that MSCs directly involve and support the formation of various tumors. As a balanced review, the tumorigenic risk of MSCs should be included in the discussion, especially as the behaviors of MSCs alter in the psoriasis microenvironment.
Author Response
comments 1.Tables 1 and 2 are not available for review.
Response 1: We apologize for the mistake. Table 1 and Table 2 were uploaded separately as attachments and we did not notice that they did not show in the manuscript. For better reading, we have embedded Table 1 and Table 2 into the main text in the revision.
comments 2:The article does not define multiple abbreviations, although they may be familiar to experts in the area.
Response 2: Thank you for pointing this out. Per suggestion, we have checked the manuscript and spelt out the full name of the abbreviations at their first appearance in the revision.
comments 3:Figures 1 and 2 seem to have been prepared using a third-party platform. Relative copyright information should be provided.
Response 3: Figures 1 and 2 were created and illustrated purely by the authors and our team whose contributions have been acknowledged.
comments 4:The caption of Figure 2 is too simple, lacking the necessary information to guide the audience.
Response 4: Thank you for pointing this out. Per suggestion, we have elaborated the captions for all figures in the revision.
comments 5:Accumulating reports have demonstrated that MSCs directly involve and support the formation of various tumors. As a balanced review, the tumorigenic risk of MSCs should be included in the discussion, especially as the behaviors of MSCs alter in the psoriasis microenvironment.
Response 5: We thank the reviewer for this comment. Indeed, there are some publications demonstrating the appearance of mesenchymal stem cells in tumor niche and they might participate in tumor growth. However, there are multiple tissue cells within tumor niche and we don’t think it’d be fair to define all of them pro-tumor formation. Whether there is a causality in transplantation of MSCs and oncogenesis remains largely undetermined. To our best knowledge, among thousands of clinical researches, there is no one report provided unequivocal evidence of tumor formation after MSCs transplantation. Therefore, we are careful about the publications regarding this issue. Nonetheless, we have added some brief description regarding this issue in Section 5 per suggestion, despite this is not the main scope for the review.
Reviewer 3 Report
Comments and Suggestions for Authors
This is a review article for the pathogenic role and therapeutic potential of mesenchymal stem cells (MSCs) in psoriasis, which has not fully reviewed so far. By referring 127 previous papers, the authors well summarized all the pathogenesis in psoriasis, the potential of MSCs originated from various tissues, and the current situation for MSCs in clinical studies in psoriasis. This is a timely review for the possible new therapy for psoriasis. The manuscript is in general well prepared, English is good, and this review provides is with various insights. However, I have some comments, which are described below.
(1) The authors mention that there are MSCs in the dermis of psoriatic skin and MSCs are derived from various tissues. However, the authors did not mention about the origin of the MSCs in the psoriatic skin. The authors should clearly mention from which tissues (bone marrow, adipose tissue or others) the MSCs in the psoriatic skin are derived.
(2) In the figure 2, the authors should clearly mention how the abnormal MSCs in the psoriatic skin are different from MSCs in normal human skin both within the figure and in the figure legend.
(3) For all the 3 figures, the authors should explain the figures in more detail in the figure legends.
(4) Tables 1 and 2 are missing. The authors should provide with these tables.
Author Response
comments 1:The authors mention that there are MSCs in the dermis of psoriatic skin and MSCs are derived from various tissues. However, the authors did not mention about the origin of the MSCs in the psoriatic skin. The authors should clearly mention from which tissues (bone marrow, adipose tissue or others) the MSCs in the psoriatic skin are derived.
Response1 : Thank you for the comment. We apologize for the misunderstanding perhaps due to the expressions in the text. To clarify, we made adjustments in Section 3.1 of the text. Evidence suggests that the pathological MSCs were directly transformed from resident MSCs in psoriasis skin. These resident MSCs change in terms of function, phenotype, etc. during disease progression (as described in 3.2 of the text). Therefore, the pathological MSCs were originated from the dermis other than from other tissue. The MSCs we described from different sources (bone marrow, fat or other tissues) that you pointed out are actually the donors currently applied in psoriasis treatment.
We have listed out the references regarding this issue for your better reading:
1. Campanati A, Orciani M, Gorbi S, Regoli F, Di Primio R, Offidani A. Effect of biologic therapies targeting tumour necrosis factor-α on cutaneous mesenchymal stem cells in psoriasis. Br J Dermatol. 2012;167(1):68-76. doi:10.1111/j.1365-2133.2012.10900.x
2. Hou R, Yan H, Niu X, et al. Gene expression profile of dermal mesenchymal stem cells from patients with psoriasis. J Eur Acad Dermatol Venereol. 2014;28(12):1782-1791. doi:10.1111/jdv.12420
3. Liu R, Yang Y, Yan X, Zhang K. Abnormalities in cytokine secretion from mesenchymal stem cells in psoriatic skin lesions. Eur J Dermatol. 2013;23(5):600-607. doi:10.1684/ejd.2013.2149
4. Han Q, Niu X, Hou R, et al. Dermal mesenchymal stem cells promoted adhesion and migration of endothelial cells by integrin in psoriasis. Cell Biol Int. 2021;45(2):358-367. doi:10.1002/cbin.11492
comments 2: In the figure 2, the authors should clearly mention how the abnormal MSCs in the psoriatic skin are different from MSCs in normal human skin both within the figure and in the figure legend.
Response 2: Thank you for the suggestion. The functions of normal and pathological MSCs were depicted in Figure 1 and 2, respectively. Per suggestion, for better reading, we have modified pathological MSCs in Figure 2 and rewritten legends for the figures in the revision.
comments 3:For all the 3 figures, the authors should explain the figures in more detail in the figure legends.
Response 3: Thank you for the suggestion. Elaborated figure legends for all 3 figures were provided in the revised manuscript l.
comments 4:Tables 1 and 2 are missing. The authors should provide with these tables.
Response 4: We apologize for the mistake. Table 1 and Table 2 were uploaded separately as attachments and we did not notice that they did not show in the manuscript. For better reading, we have embedded Table 1 and Table 2 into the main text in the revision.